# Optimizing Anthocyanin-Rich Black Cane (*Saccharum sinensis* Robx.) Silage for Ruminants Using Molasses and Iron Sulphate: A Sustainable Alternative

**Ngo Thi Minh Suong [1,2], Siwaporn Paengkoum [3,*], Rayudika Aprilia Patindra Purba [1,*] and Pramote Paengkoum [1]**

[1] School of Animal Technology and Innovation, Institute of Agricultural Technology, Suranaree University of Technology, Muang, Nakhon Ratchasima 30000, Thailand; ntmsuong@ctu.edu.vn (N.T.M.S.); pramote@sut.ac.th (P.P.)
[2] School of Animal Sciences, Agriculture Department, Can Tho University, Can Tho City 92000, Vietnam
[3] Program in Agriculture, Faculty of Science and Technology, Nakhon Ratchasima Rajabhat University, Muang, Nakhon Ratchasima 30000, Thailand
[*] Correspondence: siwaporn.p@nrru.ac.th (S.P.); rayudikaapp.007@sut.ac.th or rayudikaapp.007@gmail.com (R.A.P.P.)

**Abstract:** Anthocyanin-rich black cane (aBC) is a grass rich in lignin and carbohydrates, with an abundance of anthocyanins. Silages of aBC produced with molasses (MS) and/or ferrous sulphate (FS) mixtures may have beneficial effects on silage quality and animal performance in ruminants. However, the addition of MS and FS to ensiled grass is relatively unexplored. Therefore, this study systematically evaluated the effect of their administration at different doses to select an effective treatment to modulate the ensiling characteristics of aBC. In the first trial, fresh or pre-ensiled materials (PBC) were compared with ensiled PBC treated with: 0% MS 0% FS, 4% MS, 8% MS, 0.015% FS, 0.030% FS, 4% MS + 0.015% FS, 4% MS + 0.030% FS, 8% MS + 0.015% FS, and 8% MS + 0.030% FS on a fresh matter basis. The quality of ensiling characteristics was determined in laboratory-scale silos after 42 d of preservation. Based on these results, the second trial was further conducted in rumen cultures to ensure that the selected treatment would not impair rumen fermentation. For this, ruminal biogases, rumen fermentation profiles, and microbial communities were evaluated. Ensiled PBC with the incremental addition of MS and FS resulted in the observations for anthocyanin contents and the ensiling characteristics of the aBC silages. The combination of MS (4%) and FS (0.030%) incorporated into silages had the highest silage production effect among the experimental treatments. This combination demonstrated the sustainable mitigation of the ruminal biogases of methane and carbon dioxide without impairment of total gas production. Concurrently, this combination improved total volatile fatty acid concentrations, modulated cellulolytic bacteria, and suppressed methanogenic bacteria in rumen fluids. The results presented here indicated that addition of a mixture of 4% MS and 0.030% FS to aBC resulted in an optimal balance of ensiling characteristics and is suitable for use in ruminants.

**Keywords:** anthocyanin; biogas; fermentation; molasses; ferrous sulphate; ruminant

## 1. Introduction

A noble cane, anthocyanin-rich black cane (aBC; *Saccharum sinensis* Robx.) is a complex interspecific hybrid between *Saccharum spontaneum* and *Saccharum officinarum*, a strong-growing species of grass (Poaceae) native to mainland South Asia [1]. Although it has harder stalks than sugarcane (*S. officinarum*), aBC was generally used for sugar production due to its high sucrose content. In recent years, in Thailand, aBC has been recognized for its rapid growth, high fiber content, and abundance of anthocyanin. Despite the availability of energy and protein content, all of these are strong reasons to cultivate aBC and

consider it as a potential source of roughage supplementation for ruminant animals. The functional properties of anthocyanin as a beneficial coloring compound have been explored in livestock; it modulates the alleviation of heat stress, accelerates antioxidant activity, and stimulates the rumen microbiome [2–4]. Therefore, it has a potential to improve rumen-derived products, such as milk and meat. However, feedback from farmers indicates that feeding aBC alone could not completely meet the production requirements of ruminants because of the poor palatability of anthocyanin and low dry matter intake. Typically, plant anthocyanin has a bitter taste, and sinensis species also have a high content of lignocellulosic components [5]. This has a superior effect on the non-structural carbohydrate and degrades the structural carbohydrate during rumen digestion [6]. Therefore, most farmers do not utilize or treat it. Thus, it is left in the field to decompose or is burned in the open, leading to significant environmental impacts. Given that such agricultural activities pose a significant problem in terms of disposal, ensiling has been proposed as an effective method of preserving roughage; however, it necessitates a suitable effort prior to ensiling because biomass breakdown during anaerobic fermentation results in functional microorganisms and chemicals that reduce nutrition loss as well as harmful substances.

Several previous studies [7–9] have indicated that the pretreatment of lignocellulose agricultural biomass or materials prior to ensiling with inorganic salts is a promising tool to recover the nutritional value of highly lignified materials. Inorganic salts such as $Fe_2(SO_4)_3$ or ferrous (iron) sulphate (FS) could be used as catalysts to increase the hydrolytic efficiency of hemicellulose and cellulose in biomass during fermentation [7]. Here, the beneficial properties of FS as a source of iron and as a means of improving enzymatic hydrolysis and fermentation were primarily studied based on biomass biorefining. Nevertheless, Suong et al. [10] found that FS heptahydrate could be used for the breakdown of highly lignified black cane and for preserving abundant anthocyanins but that inclusion of FS heptahydrate without additives in black cane silages produced a slight reduction in dry matter (DM) content after 21 d of ensiling. A consensus exists that a 21-day ensiling period is insufficient to achieve a constant silage quality [11]. In a practical sense, tropical grass can be ensiled at least once per month in order to achieve quality consistency in the silage [12]. These results support the idea that ensiling aBC with iron (F) may accelerate biomass breakdown during fermentation over a longer time, which could be sufficient to ensure quality consistency in the silage.

Furthermore, the most recommended additive for ensiling low-quality roughage is molasses (MS) due to its moisture and soluble carbohydrate (WSC) content which favors the growth of anaerobic and lactic acid bacteria [13]. To the author's knowledge, the effects of FS and/or MS on the degradation of hemicellulose and cellulose sugars and the role played by silage quality on animal performance in response to dietary ensiling for ruminants have not been studied in depth. Based on this, we hypothesized that aBC ensiled with FS and/or MS mixtures for 42 d may improve the ensiling characteristics, anthocyanin stability, rumen fermentation profile, and microbial communities present in aBC silages. Therefore, this study investigated the effects of MS and FS in different doses with the aim of selecting an effective treatment to modulate the ensiling characteristics of aBC. Once a combined treatment at a practical dose (in terms of cost and avoidance of toxicity [14]) was selected, a secondary aim, prior to in vivo testing, was to check whether it would negatively affect the rumen fermentation profile, at least in in vitro studies (Figure 1). The findings of this study could provide insights into the nature of hydrolytic pretreatments for lignocellulose agricultural biomass and a roughage-based strategy for ruminant feedstuff development with positive impacts on the environment.

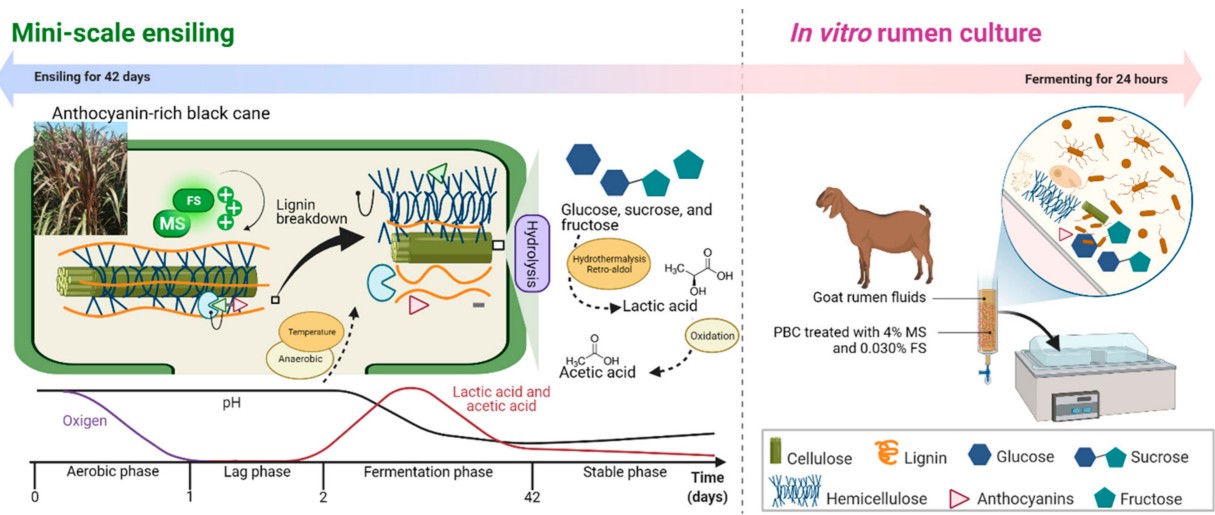

**Figure 1.** Silages of anthocyanin-rich black cane prepared with molasses (MS) and/or ferrous sulphate (FS) combinations improve silage quality and ruminant performance.

## 2. Materials and Methods

A preserved trial (Experiment 1) was conducted in laboratory-scale silos to determine the effects of MS and FS in different doses on silage production. Based on the results obtained in this mini-scale ensiling, an in vitro study (Experiment 2) was further conducted in rumen cultures to test the effect of a selected treatment and determine the ideal dose of MS and FS for rumen fermentation and bacterial communities. All experimental procedures used in this study were approved by the Animal Ethics Committee of Suranaree University of Technology (SUT 4/2558). All goats used as rumen donors were housed at the Suranaree University of Technology (SUT) goat and sheep research farm, Nakhon Ratchasima, Thailand; the animal health management was in accordance with our previous study [15]. The research was carried out in accordance with the regulations on animal experimentation and the Guidelines for the Use of Animals in Research as recommended by the National Research Council of Thailand (U1-02632-2559).

### 2.1. Roughage Harvesting and Mini-Scale Ensiling (Experiment 1)

Anthocyanin-rich black cane (aBC) was cultivated and grown in an area of 600 m$^2$ with a set plot size (50 × 50 cm) at the Suranaree University of Technology (SUT) goat and sheep research farm, Nakhon Ratchasima Province, Thailand (14°52′49.1″ N, 102°00′14.9″ E, at an elevation of 243 m above sea level). The field experiment was undertaken from August 2018 to February 2019, during the monsoon. Forage management and fertilizer use were as described in a previous study [16]. Thus, the plots were ploughed and harrowed before they were ready to use and had a basal dressing fertilizer (N-P$_2$O$_5$-K$_2$O, 50:50:50; Hydro Thai Limited, Bang Kruai, Thailand). Fresh aBC (FaBC), was sampled from random spots (n = 6) on the 60th d following 120 d of regrowth by cutting far from the soil surface (±10 cm above ground level [3]). FaBC was then mechanically chopped to an approximate length of 2–3 cm using a crop cutter and mixed well. It was then divided into two groups: pre-ensiled and silage materials. One thousand-gram portions of chopped FaBC were kept in sealed plastic bags and stored at −20 °C (pending the experiment); this was used as the control (untreated or pre-ensiled material; PBC). PBC for use as the ensiling grass was mixed homogenously and treated with ensiled combination treatments in three MS doses (0, 4, and 8%) and three FS doses (0, 0.015, and 0.030%): 0% MS 0% FS (NA), 4% MS, 8% MS, 0.015% FS, 0.030% FS, 4% MS + 0.015% FS, 4% MS + 0.030% FS, 8% MS + 0.015% FS, and 8% MS + 0.030% FS on a fresh matter (FM) basis. The experimental design was based on a 2 × 3 factorial arrangement in a completely randomized design (CRD) with six replicates. The dosage level of additives was selected on the basis of previous studies, so that it was considered a modulated ensiling characteristic, resulting

in well-preserved silage [17] and was safe for use in animals [14,18]. A portion of FS was dissolved in water (1 L) and subsequently mixed with a portion of MS. A spraying technique was then employed for adding the additives to 1 kg of pre-ensiled grass followed by mixing to homogeneity. Each fixed treatment was then transferred and placed into a silo nylon bag (Hiryu KN type, Asahi Kasei Pax Corp., Tokyo, Japan) which was then sealed with a vacuum sealer (Fresh World, TVS-2013S, Supreme Lines Co., Ltd., Bangkok, Thailand). Each treatment was performed in six silos. The ensiling process was performed in the dark at an ambient temperature of 15–25 °C for 42 d. At silo opening, the treated silages were sampled and mixed homogenously; subsamples were taken for evaluation of chemical and anthocyanin compositions, microorganisms, and fermentation quality.

*2.2. Animal Management and In Vitro Rumen Culture (Experiment 2)*

Once the combined treatment of 4% MS and 0.03% FS added to PBC was selected, an in vitro rumen culture was performed following the Hohenheim gas test method to test the effect of selected treatments (dried forms: PBC vs. NA vs. PBC treated with 4% MS + 0.030% FS) on rumen fermentation and bacterial communities, according to Menke and Steingass [19]), with a slight modification in the evaluation of total gas production and the partitioning of gas for methane and carbon dioxide production [20–22]. Rumen fluids were collected from crossbred Thai-native × Anglo–Nubian male goats (20.4 ± 2.0 kg of live weight) as part of Vorlaphim et al.'s study [6], 3 h before the morning feeding with the use of a stomach tube connected to a manual pump [6,23]. The goats were fed with a diet containing 47% cassava hay, 15% concentrate, 35% rice stubble, and 3% urea on a DM basis at 0700 and 1500 h and had free access to drinking water. Rumen fluid was moved to the laboratory (14°52′36″ N, 102°00′54″ E, at an elevation of above 200 m above sea level) in a pre-warmed thermal flask, then strained using a nylon membrane (400 μm; Fisher Scientific S.L., Madrid, Spain) into a 1000 mL filter flask containing anaerobic salivary buffer, mixed strained rumen fluids, and anaerobic salivary buffer (2:1, volume: volume) under $CO_2$ and kept at 39 °C. Anaerobic salivary buffer was prepared as described in the method for in vitro gas production [19]. Approximately 500 mg of dry sample was placed in a 100 mL calibrated Hohenheim glass syringe (Toitu, TOP Surgical Manufacturing Co., Ltd., Tokyo, Japan), which was incubated with 30 mL of rumen fluids–buffer mixture in a water bath shaker (39 °C) for 24 h. The treatments were tested in eighteen replicates within the incubation, and incubations were repeated thrice, with three runs performed. Three syringes as blanks (rumen fluids–buffer mixture only) at each run were prepared and included. The gas volume was recorded at 6, 12, and 24 h of incubation. Cumulative volume of gas production was calculated according to the model of Orskov and Mcdonald [24]; the equation used was:

$$y = a + b \, (1^{-e(-ct)}), \tag{1}$$

where a (mL/g DM) is gas production from the soluble fraction, b is gas production from the insoluble fraction (mL/g DM), c (/h) is the gas production rate constant for the insoluble fraction (b), t (h) is the incubation time, (a + b) (mL/g DM) is the potential gas production, and y is the gas produced at time t (mL/g DM).

During the analysis of gas yields, six replicated glass syringes for each incubation time of 6, 12, and 24 h were also sampled for methane and carbon dioxide gases, fermentation end-products, and microbial analysis. Briefly, 10 mL of gas was transferred into a disposable syringe with a 3-way stopcock for direct injection into the gas chromatography machine (Agilent 7890A, Agilent Technologies, Santa Clara, CA, USA) to measure the levels of methane and carbon dioxide. Calibration and chromatographic conditions were described in previous studies [20–22]. Once the glass plungers of the glass syringes were uncapped, pH was immediately measured using a pH meter (Oakton 700, Cole-Parmer, Vernon Hills, IL, USA). A volume of 10 mL of rumen fluids was prepared for determining ammonia nitrogen and volatile fatty acid (VFA) contents and analyzed similarly to the method used for pre-ensiled materials and silage forms that was described in Section 2.3. The remaining

content of each culture syringe was collected for microbial analysis using quantitative real-time PCR (qPCR) reactions.

### 2.3. Sampling and Laboratory Analysis

Pre-ensiled materials and silage materials at silo opening were separated into four subsamples. The first subsample was oven-dried at 55 °C for 24 h to a collected weight, air-equilibrated, and subsequently ground using a mesh size of 1 mm (Retsch SM 100 mill; Retsch Gmbh, Haan, Germany). The ground samples were analyzed for DM (AOAC#934.01), ash (AOAC#942.05), and crude protein (CP; total N × 6.25; AOAC#988.05) contents following the methods of the Association of Official Analytical Chemists [25]. Organic matter content (OM) was estimated as 100% minus ash percentage, which was obtained after incineration in a muffle furnace at 550 °C for 5 h. The method of Van Soest et al. [26] was sequentially used to determine neutral detergent fiber (NDF; with heat stable ± amylase, ash included), acid detergent fiber (ADF; ash included), and acid detergent lignin (ADL). Hemicellulose content was calculated as NDF minus ADF, whereas cellulose content was calculated as ADF minus ADL. The WSC content was estimated as described in a previous study [27] by peak detection using high-performance liquid chromatography (HPLC). In addition, the second subsample was lyophilized, ground, and extracted using 0.01 N hydrochloric acid (HCl) dissolved in 80% methanol solution at 50 °C for 24 h, and then the supernatant was collected and transferred into a 50 mL volumetric flask for the determination of anthocyanin composition by HPLC [28,29]. All determinations of chemical and anthocyanin compositions of pre-ensiled materials and silage forms at silo opening were performed in appropriate replications (n = 6).

The third subsample was used for microorganism count by preparing pre-ensiled materials and silage forms sampled 42 d after fermentation using the plate count method [30], with a slight modification. Briefly, approximately 20 g of subsample was covered with 180 mL sterilized distilled water in a 500 mL beaker, then the beakers were shaken well at 30 °C for 2–3 h in a rotary shaker at 200 rpm; the serial dilutions ($10^{-1}$ to $10^{-5}$) were performed appropriately in sterilized distilled water. A volume of 20 μL of each prepared dilution was poured and spread on agar plates, without bubbles. The plates were subsequently incubated in an anaerobic jar at 30 °C for 48 h. For microbial assessment, incubated plates consisting of de Man, Rogosa, and Sharpe (MRS) agar (Himedia, Mumbai, India) were used for counting numbers of lactic acid bacteria; incubated plates consisting of blue light broth agar (Sigma-Aldrich, Gillingham, UK) were used for counting numbers of coliform bacteria; incubated plates consisting of nutrient agar (Difco, St. Louis, MO, USA) were used for counting numbers of aerobic bacteria; and incubated plates consisting of potato dextrose agar (Titan Biotech, Delhi, India) were used for counting numbers of yeasts and molds. Techniques for analysis of colony appearance and cell morphology were used to assess the yeasts from molds and bacteria. All determinations of microorganism count on pre-ensiled materials and silage forms at silo opening were performed in appropriate replications (n = 6).

The last subsample was used for the evaluation of fermentative quality by extracting silages as described previously [31], with a slight modification. Forty-gram portions of each silage were placed into 500 mL beakers which were then filled with 200 mL of distilled water, the contents blended for 30 min at an ambient temperature of 27–28 °C. The mixtures were strained through filter papers (Whatman™ No. 1441-125, GE Healthcare Life Sciences, Marlborough, MA, USA). Subsequently, pH was determined using a portable pH meter (Oakton 700, Cole-Parmer, Vernon Hills, IL, USA). Lactic acid and VFA contents were measured using HPLC (Agilent Technologies 1260 Infinity, Santa Clara, CA, USA), according to a method described previously [32]; peak detection was compared and calculated, as described by Purba et al. [28]. Ammonia nitrogen ($NH_3$-N) was analyzed using a spectrophotometer (Varioskan-LUX multimode microplate reader, Thermo Fisher Scientific, Waltham, MA, USA), as per the method described previously [33,34]. All determinations

of the fermentation quality of silage forms at silo opening were performed in appropriate replications (n = 6).

### 2.4. DNA Extraction and Real-Time PCR Quantification

Genomic DNA was extracted from each culture sample following the procedure described by Yu and Morrison [35]. Total genomic DNA was extracted from 1 mL of homogenized rumen fluid sample using a QIAamp DNA Stool Mini Kit (Qiagen, Hilden, Germany). DNA quality was evaluated using agarose gel (1%) electrophoresis; DNA yield was quantified using the NanoDrop NanoVue spectrophotometer (GE Healthcare Bio-Sciences, Pittsburgh, PA, USA) at an absorbance ratio of 260:280. To preserve DNA, the DNA samples were eluted in appropriate dilutions (volume of nuclease-free water) and were stored at $-20$ °C until further analyses.

Relative abundances of selected primers in the genomic DNA of rumen fluids were quantified using a QuantiTect SYBR Green RT–PCR Kit (full master mix; Qiagen) with the selected primer set, in a Roche Lightcycler 480-II (Roche Applied Science, Basel, Switzerland), using the amplification and real-time PCR (qPCR) settings described earlier [36]. Selected primers (Table S1) of total bacteria, *Ruminococcus albus*, *Ruminococcus flavefaciens*, *Fibrobacter succinogenes*, *Butyrivibrio fibrisolvens*, *Megasphaera elsdenii*, *Streptococus bovis,* methanogen, and protozoa were purchased (Vivantis Technologies Sdn Bhd, Selangor Darul Ehsan, Malaysia) for use as references. A six-fold serial dilution of pooled DNA was prepared to create a standard curve prior to qPCR assays. To limit variations, the qPCR assay for each selected species or group of microbes was performed in triplicate for both the standards and the genomic DNA samples. Cycle threshold (Ct) values were converted into normalized relative quantities and corrected by PCR efficiency using LightCycler 480 software version 1.2.9.11. (Roche Applied Science). Population figures for each selected species or group of microbes were expressed in absolute abundances as rrs gene copies/mL of culture samples.

### 2.5. Statistical Analyses

Data for ensiling characteristics, including microorganism count, chemical composition, fermentative quality, and anthocyanin content, consisted of three levels of MS supplementation, three levels of FS supplementation, and six replications and runs, making a total of 54 observations. Data were subjected to ANOVA using the general linear model (GLM) procedure of SAS 9.4 according to a $2 \times 3$ factorial arrangement in CRD. The covariance structure was compound symmetry, which was selected in the Kolmogorov–Smirnov test of the mixed model of SAS. The average value of each treatment was computed using the least square means method (LSMEANS); the statistical model used was:

$$Y_{ij} = \mu + M_i + F_j + MF_{ij} + \varepsilon_{ij}, \tag{2}$$

where $Y_{ij}$ is an observation, $\mu$ is the overall mean, M is the MS level effect (i = 1, 2, 3), F is the FS level effect (j = 1, 2, 3), MF is the interaction effect of the MS and FS levels, and $\varepsilon_{ij}$ is the residual effect. If F-test results were significant, single degree of freedom orthogonal contrasts were performed to determine the contrasts between factors. Duncan's New Multiple Range Test (DMRT) was used to determine the differences in the treatment means at $p < 0.05$ [37].

Since data from the three consecutive runs of incubations and data observed at 6, 12, and 24 h post-incubation, were non-significant, data for total gas accumulation, methane, carbon dioxide, rumen fermentation end-product, and microbial composition as observed at 6, 12, and 24 h post-incubation were processed as a CRD and subjected to one-way ANOVA using the MIXED procedure of SAS 9.4. The statistical model used was:

$$Y_{ij} = \mu + S_i + \varepsilon_{ij}, \tag{3}$$

where $Y_{ij}$ is the observation, $\mu$ is the overall mean, Si is the treatment effect (i = 1 to 3, PBC, NA, and 4% MS + 0.030% FS), and $\varepsilon_{ij}$ is error. The significant differences among treatment

means were assessed by means of Tukey's HSD test and the level of statistical significance was pre-set at $p < 0.05$.

## 3. Results

### 3.1. Composition of Anthocyanin-Rich Black Cane

The chemical composition of pre-ensiled materials (PBC) grown in a controlled field are shown in Table 1. The DM of aBC prior to ensiling was 15.43%, and the OM, CP, NDF, ADF, and ADL were 86.81, 7.62, 79.49, 49.01, and 5.84% DM, respectively. The hemicellulose, cellulose, and WSC contents of aBC were 304.80, 431.68, and 26.13 g/kg DM, respectively. In addition, at a slightly acidic pH, aBC was observed to have an abundant concentration of anthocyanins: 4.25% cyanidin-3-glucoside (C3G), 8.51% pelargonidin-3-glucoside (P3G), 7.24% delphinidin (Del), 12.49% peonidin-3-O-glucoside (Peo3G), 10.41% malvidin-3-O-glucoside (M3G), 14.30% cyanidin (Cya), 7.24% pelargonidin (Pel), and 35.57% malvidin (Mal) of total anthocyanin (Table 2). However, aBC seemed to have low lactic acid bacteria contents ($2.4 \times 10^4$ CFU/g FM), dominated by coliform bacteria, aerobic bacteria, and yeast (Table 3).

**Table 1.** Chemical composition of anthocyanin-rich black cane prior to ensiling (pre-ensiled materials) and its silages at 42 d after fermentation.

| Item [1] | DM (g/kg FM) | OM (g/kg DM) | CP (g/kg DM) | NDF (g/kg DM) | ADF (g/kg DM) | ADL (g/kg DM) | HC (g/kg DM) | CEL (g/kg DM) | WSC (g/kg DM) |
|---|---|---|---|---|---|---|---|---|---|
| Pre-ensiled materials (PBC) | 154.34 | 868.12 | 76.24 | 794.91 | 490.11 | 58.43 | 304.80 | 431.68 | 26.13 |
| Silage additives | | | | | | | | | |
| MS 0% FS 0% (NA) | 154.52 | 859.72 | 73.83 | 790.50 | 490.90 | 61.26 | 299.60 | 429.64 | 10.42 |
| MS 4% | 151.06 | 855.04 | 73.01 | 780.61 | 488.82 | 57.37 | 291.79 | 431.45 | 11.25 |
| MS 8% | 152.62 | 855.96 | 72.95 | 780.45 | 487.93 | 57.36 | 292.52 | 430.57 | 12.06 |
| FS 0.015% | 156.13 | 841.25 | 75.01 | 788.75 | 480.76 | 50.09 | 307.99 | 430.67 | 11.52 |
| FS 0.030% | 161.42 | 834.53 | 76.17 | 789.44 | 490.81 | 49.97 | 298.63 | 440.84 | 12.42 |
| MS 4% FS 0.015% | 162.21 | 868.72 | 76.02 | 790.64 | 489.34 | 56.53 | 301.30 | 432.81 | 21.27 |
| MS 4% FS 0.030% | 169.94 | 870.52 | 76.55 | 790.52 | 490.00 | 55.73 | 300.52 | 434.27 | 23.94 |
| MS 8% FS 0.015% | 163.64 | 869.64 | 75.16 | 789.56 | 489.74 | 58.03 | 299.82 | 431.71 | 22.04 |
| MS 8% FS 0.030% | 165.25 | 870.06 | 74.98 | 780.30 | 489.84 | 57.35 | 290.46 | 432.49 | 24.03 |
| SEM | 0.109 | 0.103 | 0.095 | 0.393 | 0.083 | 0.072 | 0.043 | 0.066 | 0.090 |
| Contrast *p*-values | | | | | | | | | |
| PBC vs. Silage additives | <0.001 | <0.001 | 0.086 | 0.148 | <0.001 | <0.001 | <0.001 | <0.001 | <0.001 |
| NA vs. MS FS | <0.001 | <0.001 | 0.089 | 0.280 | <0.001 | <0.001 | <0.001 | <0.001 | <0.001 |
| NA vs. MS | 0.219 | 0.102 | 0.780 | 0.089 | 0.481 | 0.384 | <0.001 | 0.840 | <0.001 |
| NA vs. FS | <0.001 | <0.001 | 0.419 | 0.926 | <0.001 | <0.001 | <0.001 | <0.001 | <0.001 |
| NA vs. MS FS | <0.001 | <0.001 | 0.150 | 0.567 | 0.560 | <0.001 | <0.001 | <0.001 | <0.001 |
| PBC vs. MS FS | <0.001 | 0.692 | 0.635 | 0.310 | 0.899 | <0.001 | <0.001 | <0.001 | <0.001 |
| MS 4% vs. MS 8% | 0.648 | 0.772 | 0.999 | 0.999 | 0.999 | 0.999 | 0.810 | 0.999 | 0.002 |
| FS 0.015% vs. FS 0.030% | 0.005 | <0.001 | 0.917 | 0.999 | 0.002 | 0.999 | <0.001 | 0.004 | <0.001 |

[1] DM = dry matter; OM = organic matter; CP = crude protein; NDF = neutral detergent fiber; ADF = acid detergent fiber; ADL = acid detergent lignin; HC = hemicellulose; CEL = cellulose; WSC = water-soluble carbohydrate; MS = molasses; FS = ferrous sulphate; NA = ensiled PBC only; MS 4% = PBC + 4% MS; MS 8% = PBC + 8% MS; FS 0.015% = PBC + 0.015% FS; FS 0.030% = PBC + 0.030% FS; MS 4% FS 0.015% = PBC + 4% MS + 0.015% FS; MS 4% FS 0.030% = PBC + 4% MS + 0.030% FS; MS 8% FS 0.015% = PBC + 8% MS + 0.015% FS; MS 8% FS 0.030% = PBC + 8% MS + 0.030% FS on a fresh matter basis.

**Table 2.** pH and anthocyanin concentration of anthocyanin-rich black cane prior to ensiling (pre-ensiled materials) and its silages at 42 d after fermentation.

| Item [1] | pH | Anthocyanin Content (mg/g DM) | | | | | | | | |
|---|---|---|---|---|---|---|---|---|---|---|
| | | C3G | P3G | Del | Peo3G | M3G | Cya | Pel | Mal | Total |
| Pre-ensiled materials (PBC) | 5.42 | 0.047 | 0.094 | 0.080 | 0.138 | 0.115 | 0.158 | 0.080 | 0.393 | 1.105 |

**Table 2.** *Cont.*

| Item [1] | pH | Anthocyanin Content (mg/g DM) | | | | | | | | |
|---|---|---|---|---|---|---|---|---|---|---|
| | | C3G | P3G | Del | Peo3G | M3G | Cya | Pel | Mal | Total |
| Silage additives | | | | | | | | | | |
| MS 0% FS 0% (NA) | 4.76 | 0.014 | 0.021 | 0.049 | 0.047 | 0.041 | 0.074 | 0.005 | 0.040 | 0.291 |
| MS 4% | 4.17 | 0.016 | 0.034 | 0.055 | 0.063 | 0.058 | 0.045 | 0.006 | 0.064 | 0.341 |
| MS 8% | 3.74 | 0.018 | 0.037 | 0.057 | 0.064 | 0.071 | 0.057 | 0.009 | 0.077 | 0.390 |
| FS 0.015% | 4.68 | 0.015 | 0.026 | 0.061 | 0.059 | 0.080 | 0.077 | 0.005 | 0.082 | 0.405 |
| FS 0.030% | 4.21 | 0.015 | 0.027 | 0.072 | 0.072 | 0.089 | 0.081 | 0.007 | 0.119 | 0.482 |
| MS 4% FS 0.015% | 3.76 | 0.020 | 0.039 | 0.085 | 0.126 | 0.097 | 0.174 | 0.008 | 0.122 | 0.671 |
| MS 4% FS 0.030% | 3.71 | 0.022 | 0.041 | 0.098 | 0.133 | 0.129 | 0.187 | 0.008 | 0.129 | 0.747 |
| MS 8% FS 0.015% | 3.72 | 0.020 | 0.045 | 0.113 | 0.13 | 0.138 | 0.172 | 0.010 | 0.132 | 0.760 |
| MS 8% FS 0.030% | 3.69 | 0.022 | 0.049 | 0.136 | 0.142 | 0.144 | 0.185 | 0.009 | 0.138 | 0.825 |
| SEM | 0.036 | 0.002 | 0.007 | 0.014 | 0.016 | 0.022 | 0.019 | 0.004 | 0.017 | 0.038 |
| Contrast *p*-values | | | | | | | | | | |
| PBC vs. Silage additives | <0.001 | <0.001 | <0.001 | 0.001 | <0.001 | 0.020 | <0.001 | <0.001 | <0.001 | <0.001 |
| NA vs. MS FS | <0.001 | 0.131 | 0.095 | 0.004 | <0.001 | 0.015 | <0.001 | 0.158 | <0.001 | <0.001 |
| NA vs. MS | <0.001 | 0.414 | 0.241 | 0.898 | 0.472 | 0.436 | 0.187 | 0.163 | 0.108 | 0.002 |
| NA vs. FS | 0.002 | 0.946 | 0.630 | 0.236 | 0.480 | 0.191 | 0.971 | 0.361 | <0.001 | <0.001 |
| NA vs. MS FS | <0.001 | 0.208 | 0.054 | 0.003 | 0.003 | 0.034 | 0.006 | 0.270 | <0.001 | <0.001 |
| PBC vs. MS FS | <0.001 | <0.001 | <0.001 | 0.160 | 0.979 | 0.708 | 0.896 | 0.003 | <0.001 | 0.007 |
| MS 4% vs. MS 8% | <0.001 | 0.740 | 0.999 | 0.999 | 0.999 | 0.999 | 0.999 | 0.249 | 0.763 | 0.044 |
| FS 0.015% vs. FS 0.030% | <0.001 | 0.999 | 0.999 | 0.999 | 0.999 | 0.999 | 0.999 | 0.596 | 0.038 | 0.002 |

[1] C3G = cyanidin-3-glucoside; P3G = pelargonidin-3-glucoside; Del = delphinidin; Peo3G = peonidin-3-O-glucoside; M3G = malvidin-3-O-glucoside; Cya = cyanidin; Pel = pelargonidin; Mal = malvidin; MS = molasses; FS = ferrous sulphate; NA = ensiled PBC only; MS 4% = PBC + 4% MS; MS 8% = PBC + 8% MS; FS 0.015% = PBC + 0.015% FS; FS 0.030% = PBC + 0.030% FS; MS 4% FS 0.015% = PBC + 4% MS + 0.015% FS; MS 4% FS 0.030% = PBC + 4% MS + 0.030% FS; MS 8% FS 0.015% = PBC + 8% MS + 0.015% FS; MS 8% FS 0.030% = PBC + 8% MS + 0.030% FS on a fresh matter basis.

**Table 3.** Microorganism counts of anthocyanin-rich black cane prior to ensiling (pre-ensiled materials) and its silages at 42 d after fermentation.

| Item [1] | Microorganism (CFU/g Fresh Matter) | | | | |
|---|---|---|---|---|---|
| | Lactic Acid Bacteria | Coliform Bacteria | Aerobic Bacteria | Yeasts | Molds |
| Pre-ensiled materials (PBC) | $2.4 \times 10^4$ | $2.2 \times 10^7$ | $4.5 \times 10^6$ | $2.6 \times 10^8$ | nd |
| Silage additives | | | | | |
| MS 0% FS 0% (NA) | $5.3 \times 10^7$ | $2.5 \times 10^4$ | $8.8 \times 10^5$ | $4.1 \times 10^6$ | nd |
| MS 4% | $1.2 \times 10^8$ | $2.3 \times 10^4$ | $7.5 \times 10^4$ | nd | nd |
| MS 8% | $3.5 \times 10^8$ | $2.1 \times 10^4$ | $3.3 \times 10^4$ | nd | nd |
| FS 0.015% | $8.4 \times 10^7$ | $2.2 \times 10^4$ | $2.0 \times 10^5$ | nd | nd |
| FS 0.030% | $1.1 \times 10^8$ | $2.1 \times 10^4$ | $6.3 \times 10^4$ | nd | nd |
| MS 4% FS 0.015% | $2.4 \times 10^8$ | $1.8 \times 10^3$ | $4.1 \times 10^4$ | nd | nd |
| MS 4% FS 0.030% | $5.4 \times 10^8$ | $1.3 \times 10^3$ | $3.1 \times 10^4$ | nd | nd |
| MS 8% FS 0.015% | $3.9 \times 10^8$ | $1.7 \times 10^3$ | $3.1 \times 10^4$ | nd | nd |
| MS 8% FS 0.030% | $5.5 \times 10^8$ | $1.3 \times 10^3$ | $3.0 \times 10^4$ | nd | nd |
| SEM | 2.032 | 0.980 | 0.878 | 0.430 | 0.000 |
| Contrast *p*-values | | | | | |
| PBC vs. Silage additives | <0.001 | <0.001 | <0.001 | <0.001 | 0.000 |
| NA vs. MS FS | <0.001 | <0.001 | <0.001 | 0.000 | 0.000 |
| NA vs. MS | <0.001 | <0.001 | <0.001 | 0.000 | 0.000 |
| NA vs. FS | <0.001 | <0.001 | <0.001 | 0.000 | 0.000 |
| NA vs. MS FS | <0.001 | <0.001 | <0.001 | 0.000 | 0.000 |
| PBC vs. MS FS | <0.001 | <0.001 | <0.001 | 0.000 | 0.000 |

**Table 3.** *Cont.*

| Item [1] | Microorganism (CFU/g Fresh Matter) | | | | |
|---|---|---|---|---|---|
| | Lactic Acid Bacteria | Coliform Bacteria | Aerobic Bacteria | Yeasts | Molds |
| MS 4% vs. MS 8% | <0.001 | <0.001 | <0.001 | 0.000 | 0.000 |
| FS 0.015% vs. FS 0.030% | <0.001 | <0.001 | <0.001 | 0.000 | 0.000 |

[1] nd = not detected; MS = molasses; FS = ferrous sulphate; NA = ensiled PBC only; MS 4% = PBC + 4% MS; MS 8% = PBC + 8% MS; FS 0.015% = PBC + 0.015% FS; FS 0.030% = PBC + 0.030% FS; MS 4% FS 0.015% = PBC + 4% MS + 0.015% FS; MS 4% FS 0.030% = PBC + 4% MS + 0.030% FS; MS 8% FS 0.015% = PBC + 8% MS + 0.015% FS; MS 8% FS 0.030% = PBC + 8% MS + 0.030% FS on a fresh matter basis.

*3.2. Chemical Composition, Anthocyanin Content, and Ensiling Characteristics of Anthocyanin-Rich Black Cane*

After 42 d of ensiling, the chemical compositions of pre-ensiled materials were affected ($p < 0.001$) by the addition of experimental additives (Table 1). The DM content of ensiled aBC (silage forms) was significantly greater ($p < 0.001$) than pre-ensiled materials, ranging from 154.52 to 169.94 g/kg; the increase was due to the addition of FS ($p = 0.005$). Ensiling aBC with the addition of MS did not modify OM, CP, NDF, or ADF contents, but the addition of MS with FS increased biodegradable lignocellulose contents ($p < 0.001$). The combination of MS and FS increased hemicellulose, cellulose, and WSC contents in aBC during ensiling ($p < 0.001$). These combinations were also observed to maintain the OM and CP contents at levels comparable to those of fresh aBC. However, increasing levels of either MS (0 vs. 4 vs. 8%) or FS (0 vs. 0.015 vs. 0.030%) resulted in a shift in the degraded organic contents of aBC during ensiling. Among experimental additives, PBC + 4% MS + 0.030% FS was considered to be more pronounced for testing in in vitro gas production due to its preserved chemical composition.

In addition, the anthocyanin content of pre-ensiled materials was affected ($p < 0.001$) by adding experimental additives during ensiling for 42 d (Table 2). A significant reduction ($p < 0.05$) in anthocyanin preservation was observed with MS addition (up to 71% reduction), FS addition (up to 64% reduction), and MS–FS addition (up to 32% reduction) as compared with pre-ensiled materials. Increasing levels of either MS (0 vs. 4 vs. 8%) or FS (0 vs. 0.015 vs. 0.030%) was observed to decrease the degraded anthocyanin content of aBC during ensiling. Among experimental additives, however, PBC + 4% MS + 0.030% FS seemed to be more efficient in reducing the loss of anthocyanin content. All experimental additives were observed to decrease the pH of aBC during ensiling ($p < 0.001$; Table 2). Among the anthocyanin contents of aBC during ensiling, pH-increasing anthocyanin modulated the preservation of delphinidin, peonidin-3-O-glucoside, malvidin-3-O-glucoside, and cyanidin.

The addition of experimental additives improved the ensiling characteristics of aBC, including microbial count (Table 3). MS and FS, both alone and in combination, increased the presence of lactic acid bacteria by approximately four-fold compared with fresh aBC or pre-ensiled materials ($p < 0.001$). Adding MS decreased the presence of coliform bacteria in the range of $1.3 \times 10^3$–$2.3 \times 10^4$ CFU/g FM ($p < 0.001$) and aerobic bacteria in the range of $3.0 \times 10^3$–$2.0 \times 10^5$ CFU/g FM ($p < 0.001$). Yeast and molds were not detected in all silages treated with MS and FS (Table 3). Moreover, fermentation quality data for aBC silage after 42 d of ensiling are presented in Table 4. The addition of MS and FS did not affect the concentration of ammonia nitrogen but increased lactic acid and acetic acid levels in all silages ($p < 0.001$). As expected, propionic acid and butyric acid were not detected in all silages treated with MS and FS (Table 4). Increasing the level of either MS (0 vs. 4 vs. 8%) or FS (0 vs. 0.015 vs. 0.030%) was observed to be effective in controlling microbes and fermentation profiles during silage fermentation. Among experimental additives, PBC + 4% MS + 0.030% FS has been shown to stimulate better silage production and it should be tested further to investigate rumen fermentation and bacterial communities, including the sustainable mitigation of ruminal biogases such as methane and carbon dioxide.

**Table 4.** Fermentative quality of anthocyanin-rich black cane prior to ensiling (pre-ensiled materials) and its silages at 42 d after fermentation.

| Item [1] | Organic Compound (g/kg DM) | | | | |
|---|---|---|---|---|---|
| | Ammonia Nitrogen | Lactic Acid | Acetic ACID | Propionic Acid | Butyric Acid |
| Silage additives | | | | | |
| MS 0% FS 0% (NA) | 0.31 | 35.16 | 26.71 | nd | nd |
| MS 4% | 0.29 | 49.25 | 27.54 | nd | nd |
| MS 8% | 0.21 | 68.41 | 27.37 | nd | nd |
| FS 0.015% | 0.27 | 43.37 | 28.28 | nd | nd |
| FS 0.030% | 0.25 | 48.34 | 29.66 | nd | nd |
| MS 4% FS 0.015% | 0.26 | 53.63 | 28.82 | nd | nd |
| MS 4% FS 0.030% | 0.27 | 80.47 | 28.86 | nd | nd |
| MS 8% FS 0.015% | 0.31 | 72.21 | 28.14 | nd | nd |
| MS 8% FS 0.030% | 0.28 | 83.29 | 28.73 | nd | nd |
| SEM | 0.040 | 0.196 | 0.097 | 0.000 | 0.000 |
| Contrast *p*-values | | | | | |
| NA vs. MS FS | 0.846 | <0.001 | <0.001 | 0.000 | 0.000 |
| NA vs. MS | 0.282 | <0.001 | 0.136 | 0.000 | 0.000 |
| NA vs. FS | 0.654 | <0.001 | <0.001 | 0.000 | 0.000 |
| NA vs. MS FS | 0.917 | <0.001 | <0.001 | 0.000 | 0.000 |
| MS 4% vs. MS 8% | 0.250 | <0.001 | 0.897 | 0.000 | 0.000 |
| FS 0.015% vs. FS 0.030% | 0.999 | <0.001 | <0.001 | 0.000 | 0.000 |

[1] nd = not detected; MS = molasses; FS = ferrous sulphate; NA = ensiled pre-ensiled black cane (PBC) only; MS 4% = PBC + 4% MS; MS 8% = PBC + 8% MS; FS 0.015% = PBC + 0.015% FS; FS 0.030% = PBC + 0.030% FS; MS 4% FS 0.015% = PBC + 4% MS + 0.015% FS; MS 4% FS 0.030% = PBC + 4% MS + 0.030% FS; MS 8% FS 0.015% = PBC + 8% MS + 0.015% FS; MS 8% FS 0.030% = PBC + 8% MS + 0.030% FS on a fresh matter basis.

*3.3. Effect of 4% MS + 0.030% FS Incorporated into Silages on Rumen Fermentation Profiles and Microbial Communities in Rumen Fluids*

Rumen fermentation profiles were affected ($p < 0.05$) by incubation with both PBC and ensiled forms (NA and PBC + 4% MS + 0.030% FS), except for ammonia nitrogen and pH (Table 5). The concentrations of total VFAs increased in all incubations throughout the 24 h. We determined similar values for both PBC and PBC + 4% MS + 0.030% FS at three time points (6, 12, 24 h) during in vitro rumen incubation with respect to total VFA concentrations. Compared to NA, PBC and PBC + 4% MS + 0.030% FS showed increased total VFA concentrations ($p < 0.05$). Despite a similar production of propionate and butyrate in both PBC and ensiled forms (NA and PBC + 4% MS + 0.030% FS), only PBC and PBC + 4% MS + 0.030% FS showed an increase in the proportion of acetic acid ($p < 0.05$) compared with NA throughout the 24 h of rumen incubation. This resulted in an increase in the ratio of acetic acid to propionic acid, as observed for PBC and PBC + 4% MS + 0.030% FS (Table 5).

**Table 5.** Fermentation end-products of both pre-ensiled materials and ensiled forms at three time points (6, 12, 24 h) during the in vitro rumen incubation.

| Item [1] | Time (h) | PBC | NA | MS 4% FS 0.030% | SEM | *p*-Value |
|---|---|---|---|---|---|---|
| pH | 6 | 6.84 | 6.86 | 6.55 | 0.106 | 0.391 |
| | 12 | 6.74 | 6.74 | 6.48 | 0.196 | 0.635 |
| | 24 | 6.55 | 6.71 | 6.44 | 0.271 | 0.791 |
| Ammonia nitrogen (mg/dL) | 6 | 14.22 | 14.93 | 15.86 | 0.970 | 0.666 |
| | 12 | 15.62 | 15.04 | 15.97 | 0.561 | 0.549 |
| | 24 | 15.96 | 15.67 | 16.01 | 0.286 | 0.687 |
| Total volatile fatty acids (mmol/L) | 6 | 93.16 [a] | 80.56 [b] | 91.99 [a] | 1.793 | 0.001 |
| | 12 | 105.75 [a] | 92.97 [b] | 108.16 [a] | 1.017 | <0.001 |
| | 24 | 114.88 [a] | 100.05 [b] | 115.48 [a] | 0.613 | <0.001 |

**Table 5.** *Cont.*

| Item [1] | Time (h) | PBC | NA | MS 4% FS 0.030% | SEM | *p*-Value |
|---|---|---|---|---|---|---|
| Acetic acid (mol/100 mol) | 6 | 51.01 [a] | 49.94 [b] | 51.16 [a] | 0.287 | 0.018 |
| | 12 | 51.64 [a] | 50.91 [b] | 51.48 [a] | 0.136 | 0.010 |
| | 24 | 51.58 [a] | 50.65 [b] | 51.50 [a] | 0.161 | 0.002 |
| Propionic acid (mol/100 mol) | 6 | 37.31 | 37.81 | 37.67 | 0.166 | 0.166 |
| | 12 | 35.18 | 35.85 | 35.56 | 0.256 | 0.218 |
| | 24 | 35.16 | 35.76 | 35.6 | 0.162 | 0.081 |
| Butyric acid (mol/100 mol) | 6 | 11.68 | 12.25 | 11.17 | 0.292 | 0.106 |
| | 12 | 13.18 | 13.24 | 12.96 | 0.099 | 0.373 |
| | 24 | 13.26 | 13.59 | 12.9 | 0.171 | 0.056 |
| Ratio acetic acid to propionic acid | 6 | 1.37 [a] | 1.32 [b] | 1.36 [a] | 0.007 | 0.001 |
| | 12 | 1.47 [a] | 1.42 [b] | 1.45 [a] | 0.006 | 0.003 |
| | 24 | 1.47 [a] | 1.42 [b] | 1.45 [a] | 0.005 | <0.001 |

[1] PBC = pre-ensiled materials; NA = ensiled PBC only; MS 4% FS 0.030% = PBC + 4% MS + 0.030% FS on a fresh matter basis; SEM = standard error of mean. Values within a row followed by different letters are significantly different ($p < 0.05$).

Furthermore, the microbial community in rumen fluid was affected ($p < 0.05$) by incubation of both PBC and ensiled forms (NA and PBC + 4% MS + 0.030% FS), but only remarkably affected cellulolytic bacteria and methanogenic bacteria (Table 6). We noted comparable results for both PBC and PBC + 4% MS + 0.030% FS at three time points (6, 12, 24 h) during in vitro rumen incubation, with modulations of cellulolytic bacteria (*R. albus*) contents, though the PBC and PBC + 4% MS + 0.030% FS influences were terminated after 6 h of rumen incubation. Compared to NA, the PBC and PBC + 4% MS + 0.030% FS had a greater influence with respect to increasing proportions of cellulolytic bacteria ($p < 0.05$). In general, incubation of PBC resulted in increased proportions of cellulolytic and methanogenic bacteria, though NA and PBC + 4% MS + 0.030% FS had lower proportions of methanogenic bacteria as compared to PBC at three time points (6, 12, 24 h) during in vitro rumen incubation ($p < 0.05$). Concurrently, there were no differences in the proportions of total bacteria—*F. succinogenes*, *B. fibrisolvens*, *M. elsdenii*, *S. bovis*—or protozoa for either the incubation of PBC or ensiled forms (NA and PBC + 4% MS + 0.030% FS). Based on the above observations, PBC + 4% MS + 0.030% FS was found to improve rumen fermentation profiles and changes in the microbial communities in rumen fluid.

**Table 6.** Microbial composition ($\log_{10}$ 16S rRNA gene copies/mL) of both pre-ensiled materials and ensiled forms at three time points (6, 12, 24 h) during the in vitro rumen incubation.

| Item [1] | Time (h) | PBC | NA | MS 4% FS 0.030% | SEM | *p*-Value |
|---|---|---|---|---|---|---|
| Total bacteria | 6 | 9.26 | 9.29 | 9.38 | 0.597 | 0.991 |
| | 12 | 9.74 | 9.35 | 9.51 | 0.493 | 0.885 |
| | 24 | 9.77 | 9.45 | 9.31 | 0.356 | 0.706 |
| *Ruminococcus albus* | 6 | 7.65 [a] | 7.33 [b] | 7.75 [a] | 0.028 | 0.001 |
| | 12 | 7.72 [a] | 7.45 [b] | 7.87 [a] | 0.038 | 0.007 |
| | 24 | 7.88 [a] | 7.63 [b] | 7.94 [a] | 0.016 | <0.001 |
| *Ruminococcus flavefaciens* | 6 | 5.71 [a] | 4.22 [b] | 6.60 [a] | 0.268 | 0.003 |
| | 12 | 6.62 | 5.97 | 6.86 | 0.272 | 0.135 |
| | 24 | 6.94 | 6.78 | 7.09 | 0.160 | 0.572 |
| *Fibrobacter succinogenes* | 6 | 7.91 | 7.87 | 7.70 | 0.157 | 0.674 |
| | 12 | 8.40 | 8.28 | 8.41 | 0.049 | 0.402 |
| | 24 | 8.58 | 8.39 | 8.51 | 0.047 | 0.247 |
| *Butyrivibrio fibrisolvens* | 6 | 6.72 | 6.96 | 6.94 | 0.094 | 0.311 |
| | 12 | 7.03 | 7.16 | 7.00 | 0.158 | 0.831 |
| | 24 | 6.99 | 6.89 | 7.02 | 0.087 | 0.683 |

**Table 6.** *Cont.*

| Item [1] | Time (h) | PBC | NA | MS 4% FS 0.030% | SEM | *p*-Value |
|---|---|---|---|---|---|---|
| *Megasphaera elsdenii* | 6 | 3.69 | 3.71 | 3.72 | 0.131 | 0.995 |
| | 12 | 3.97 | 3.73 | 3.71 | 0.111 | 0.535 |
| | 24 | 3.67 | 3.68 | 3.94 | 0.174 | 0.695 |
| *Streptococus bovis* | 6 | 5.60 | 4.53 | 4.66 | 0.337 | 0.153 |
| | 12 | 5.56 | 5.42 | 5.36 | 0.391 | 0.972 |
| | 24 | 5.44 | 5.58 | 5.24 | 0.204 | 0.665 |
| Methanogen | 6 | 5.77 [a] | 5.38 [b] | 5.42 [b] | 0.056 | 0.006 |
| | 12 | 6.03 [a] | 5.65 [b] | 5.72 [b] | 0.039 | 0.007 |
| | 24 | 6.95 [a] | 6.62 [b] | 6.57 [b] | 0.050 | 0.015 |
| Protozoa | 6 | 6.04 | 6.35 | 6.13 | 0.314 | 0.782 |
| | 12 | 6.05 | 6.19 | 6.21 | 0.252 | 0.933 |
| | 24 | 6.33 | 6.31 | 6.00 | 0.155 | 0.467 |

[1] PBC = pre-ensiled materials; NA = ensiled PBC only; MS 4% FS 0.030% = PBC + 4% MS + 0.030% FS on a fresh matter basis; SEM = standard error of mean. Values within a row followed by different letters are significantly different ($p < 0.05$).

### 3.4. Suitability of 4% MS + 0.030% FS Incorporated into Silages for the Sustainable Mitigation of Ruminal Biogases

To evaluate the influence of 4% MS + 0.030% FS incorporated into silages on ruminal biogases (methane and carbon dioxide), we measured total gas production, methane production, and carbon dioxide production from the in vitro rumen incubations. Compared to PBC, incubation of ensiled forms (NA) showed a greater reduction in total gas production, whereas PBC + 4% MS + 0.030% FS resulted in a trend similar to that of PBC, producing total gas for 24 h (Figure 2a). This resulted in increases in the proportions of carbon dioxide (Figure 2b) and methane (Figure 2c) produced from the incubation of PBC at three time points (6, 12, 24 h) during in vitro rumen incubation. Similarly, PBC + 4% MS + 0.030% FS showed comparable carbon dioxide production. In contrast, along with NA, incubation of PBC + 4% MS + 0.030% FS showed greater mitigation of methane production compared to PBC ($p < 0.05$) at three time points (6, 12, 24 h) during in vitro rumen incubation (Figure 2c). The sustainable mitigation of the ruminal biogases methane and carbon dioxide could be attributed to silages treated with 4% MS + 0.030% FS, without impairing total gas production.

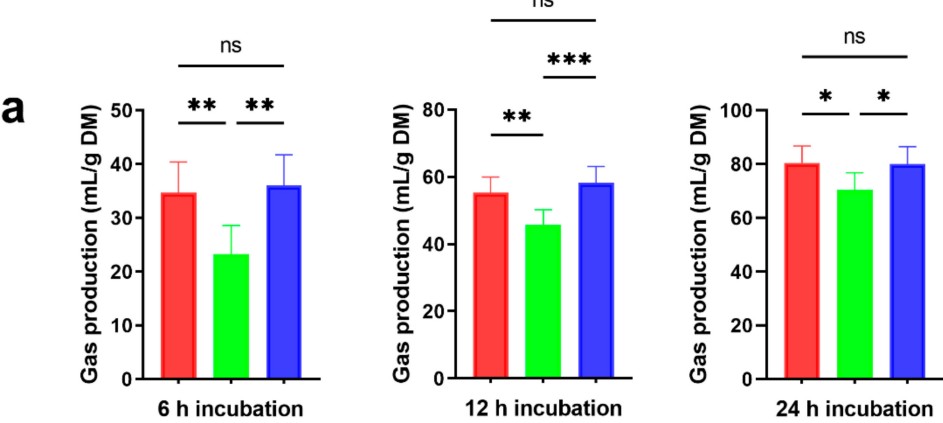

**Figure 2.** *Cont.*

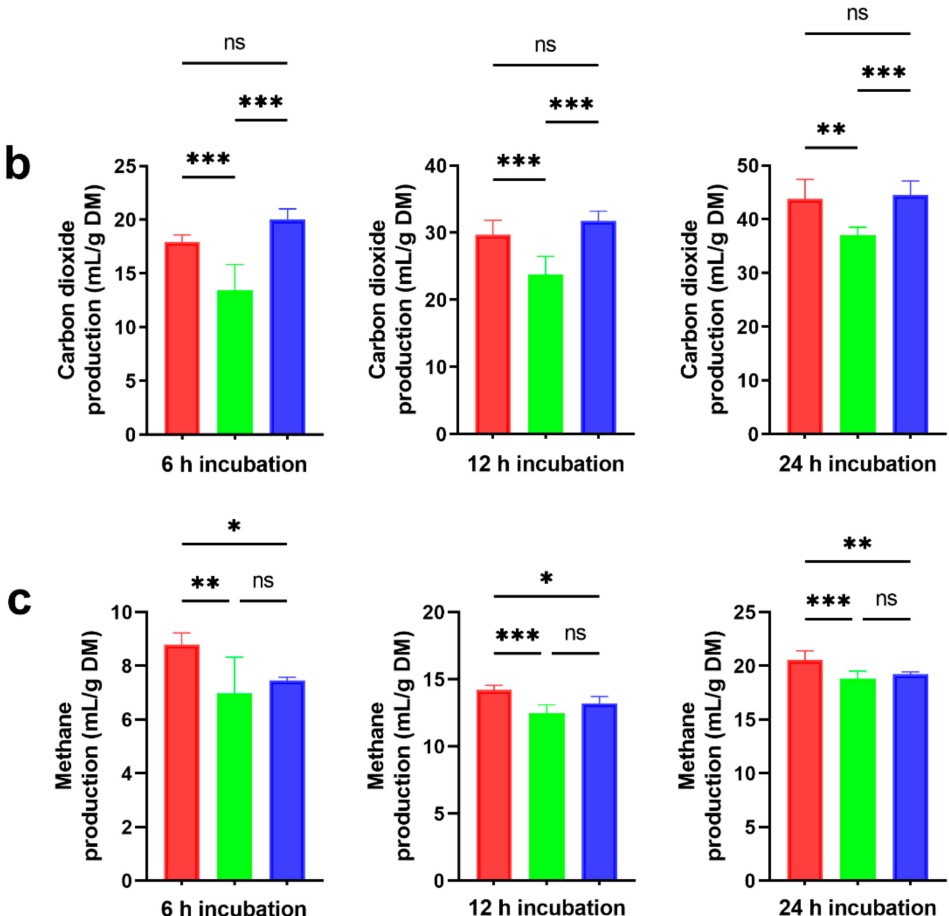

**Figure 2.** Ruminal gas production of anthocyanin-rich black cane, in both pre-ensiled materials (PBC, ■) and ensiled forms (NA, ■; 4% MS + 0.030% FS, ■), at three time points (6, 12, 24 h) during in vitro rumen incubation: (**a**) total gas production; (**b**) carbon dioxide production; (**c**) methane production. The values shown are the means, with standard errors represented by vertical bars. Significantly different values are indicated as follows: *, $p < 0.05$; **, $p < 0.01$; ***, $p < 0.001$; ns, $p > 0.05$.

## 4. Discussion

### 4.1. Anthocyanin-Rich Black Cane Has a High Fiber Content and Abundance of Anthocyanin, as Well as High Lignification or Silicification of Lignocellulosic Biomass

The analysis of the chemical composition of forage or roughage as potential ruminant feedstuff is essential prior to its use in agricultural biomass because of the heterogeneity of its anatomical structure and chemical properties. Differences in the chemical compositions of plant biomasses are dependent on biological and geographical factors, including plant sampling, plant age, and location [38]. In the present study, aBC was observed to have an abundance of anthocyanin. The values for total anthocyanin and anthocyanin contents were similar to those observed for Chinese sugarcane [39,40], but specific anthocyanin contents, such as cyanidin-3-glucoside (C3G), pelargonidin-3-glucoside (P3G), delphinidin (Del), peonidin-3-O-glucoside (Peo3G), malvidin-3-O-glucoside (M3G), cyanidin (Cya), pelargonidin (Pel), and malvidin (Mal) were higher in aBC. However, the major chemical composition of *S. sinensis* has not yet been reported. The fiber content of aBC appears to be higher when compared to other tropical roughages or grasses, which could make it a beneficial dietary carbohydrate-rich feed for ruminant animals [41]. It should be noted that aBC in the present study was observed to have high lignification or silicification of lignocellulosic biomass, resulting in poor nutritional value. Therefore, pretreatment of lignocellulose is recommended as a method to recover the nutritional value of aBC [7–9], and this can be extrapolated, at least qualitatively, to practical feeding conditions [6]. Silage

production using the proposed aBC processing might also provide a method to remedy some of the existing environmental impacts. In addition, the values of DM, OM, CP, NDF, ADF, ADL, HC, CEL, and WSC of aBC in the present study are generally sufficient for further roughage ensiling. This assessment was due to the moderately high ensilability index (EI; +12) of FaBC, which was used to predict the ensilage process [42]. The numbers of FaBC microbes may also have to be controlled during silage fermentation by using the proposed experimental additives, including MS and FS, either alone or in combination.

*4.2. Combination of Molasses and Ferrous Sulphate Improved Chemical Composition, Anthocyanin Content, and Ensiling Characteristics of Anthocyanin-Rich Black Cane*

The successful production of silage treated with MS and FS is based on the stability of ensiled materials, with a greater recovery of DM, energy, and highly digestible nutrients compared with untreated silages [11]. This could be achieved by controlled moisture shifts during silage fermentation. Previously, it was speculated that moisture content could explain whether an excellent, average, or poor fermentation occurred—excessive moisture content resulted in a greater loss of DM content and led to low-quality silage [11]. Although the use of FS for grass silage is uncommon, the use of preservatives is often preferred over the use of MS due to their ability to reduce the loss of chemical characteristics of ensiled materials during the process of preservation by means of bringing about a reduction in both respiration and fermentation [13]. However, among experimental additives, the addition of MS alone is less likely to induce the chemical composition of aBC, and a combination of MS and FS is more likely to enhance anaerobic fermentation and breakdown of lignocellulosic components. It has been shown that FS can be used as a catalyst by increasing the degradation of hemicellulose and cellulose sugars in corn stover [7]. The greater hydrolysis efficiencies of hemicellulose and cellulose in lignocellulose agricultural biomass during biomass biorefining yielded higher soluble carbohydrate contents. Generally, hemicellulose is easier to degrade than cellulose during ensiling. However, FS was observed to successfully degrade both hemicellulose and cellulose in aBC, leading to higher contents of non-structural and structural carbohydrates. Therefore, the reason for WSC increase as a consequence of FS addition in our study could be attributed to an increase in the breakdown of lignocellulosic components [5], resulting in superior effects on the non-structural carbohydrate and degradation of the structural carbohydrate. It is possible that the increase in WSC of aBC silage resulted in an increased recovery of DM and energy because of the use of WSC (e.g., glucose) by homolactic acid bacteria for growth and production of lactic acid during fermentation [43].

To the best of our knowledge, this is the first study to report the synergistic effect of MS and FS on fermentation quality and microbial composition. In an ideal fermentation process, the most common measurements used for evaluating silage fermentation include pH, concentrations of organic acids and ammonia nitrogen, and the variation of ensiled microbial populations [11]. In the present study, we demonstrated that a combination of MS and FS facilitated the silage quality of aBC through the following fermentation characteristics. The pH of additive-treated silages was close to 3.7, ensuring good preservation of aBC silages [11]. Other studies reported that ensiled sugarcane (*S. officinarum)* had a higher pH (4.5–4.7) [13,27]. Our results indicate that the decline in pH was due to the presence of FS during the aBC ensiling process and that, together with MS, it created the more acidic conditions. The reduced pH may be related to the concentration of buffering capacity and lactic acid [11]. Lactic acid in silage is the dominant product from the preservation of crops or grasses as a result of lactic acid bacteria being able to exert metabolic water availability for growth. In the present study, the combination of MS and FS enhanced bacterial growth, which resulted in a tripling of the production of lactic acid as compared to ensiled aBC alone and a doubling of the production of lactic acid as compared to ensiled aBC with MS. Similar to lactic acid, the concentration of acetic acid in the present study seems to increase in response to the ensilage of aBC with combined MS and FS. Our findings showed that the levels of acetic acid were moderate (up to 3%, treated with 4% MS and 0.030% FS), indicat-

ing that yeasts were undetected in our silages after preserving aBC with a combination of MS and FS. Previously, it was discussed systematically that moderate concentrations (3–4%) of acetic acid in silage have the beneficial property of inhibiting yeasts during fermentation, thereby improving the stability of silages when silages are exposed to air [10,11]. Our proposed experimental additives using combined MS and FS resulted in well-fermented silages. Notably, well-fermented silages have very low concentrations (<0.1%) of propionic acid and butyric acid. Clostridial organisms are responsible for the metabolism of soluble sugars or organic (lactic) acids to yield butyric acid which leads to large losses of DM and poor recovery of energy [43]. In fact, in the present study, preserving aBC with combined MS and FS has been shown to suppress the growth of coliform bacteria, suggesting a superior effect for *Clostridium* in silages. The limited growth of *Clostridium* in silages might explain why some silages observed in the present study had undetected levels of propionic acid and butyric acid [11]. FS has been shown to inhibit unwanted organic contaminants and might be related to sulphate-reducing bacteria, including *Clostridium*, through oxidation reactions that involve ferrous ions ($Fe^{2+}$) and hydrogen peroxide ($H_2O_2$) [8]. Our results suggest that the use of MS and FS for ensiling lignocellulose agricultural biomass is fairly important because they could enhance fermentative quality without significant reductions in chemical composition and stimulate the growth of desirable ensiling bacteria by suppressing undesirable ensiling bacteria. The most likely reason for this is that FS might deteriorate those undesirable ensiling bacteria by undertaking cell wall synthesis or nucleic acid synthesis. However, the mechanism of these actions remains unclear and further investigations are needed to clarify the role of FS in the presence of undesirable ensiling bacteria.

To accurately provide anthocyanin to ruminant animals by feeding ensiled aBC, it is essential to investigate the quantitative changes in anthocyanin contents during preservation. Generally, anthocyanin and other major compounds, including vitamins and carbohydrates, are used by lactic acid bacteria for growth and produce only lactic acid. Anthocyanin stability during preservation is extremely responsive to environmental conditions, including high pH, oxygen, high temperature, and light [44]. Temperature and light could be excluded as factors in this study because all ensiled aBC samples were compressed under anaerobic conditions, placed in laboratory-scale silos, and kept in a dark environment at an ambient temperature of 15–25 °C. Interestingly, some previous studies [31,44] concluded that the degradation rates of anthocyanins were positively correlated with pH values under similar conditions of ensilage. The results of the present study demonstrated that all silages had a lower pH value after 42 d of ensilage, which facilitated the creation of an environment conducive to the stability of anthocyanins. Therefore, the decreased range of anthocyanins in ensiled aBC treated with MS alone was greater compared to ensiled aBC treated with a combination of MS and FS. This suggests that the declines in pH in MS–FS-treated-silages were regulated by FS. Another reason might be that, as was previously mentioned, FS could enhance the breakdown of lignin fractions in aBC, resulting in the availability of more sugar, thereby promoting ensilage. The present results are in agreement with those of previous studies, in which approximately 42% of anthocyanin was preserved in colored barley [32] and approximately 52% was preserved in anthocyanin-rich purple corn stover silage [31]. Moreover, our results indicate that pH-increasing anthocyanins seemed to have superior effects on the delphinidin, peonidin-3-O-glucoside, malvidin-3-O-glucoside, and cyanidin contents of aBC, which is consistent with the findings of previous studies [31,45,46].

### 4.3. Combination of Molasses and Ferrous Sulphate Increases Total VFA Concentrations, Modulates Cellulolytic Bacteria, and Suppresses Methanogenic Bacteria in Rumen Fluid

Similar to previous studies [2,47] that observed purple corn anthocyanin, our results suggest that there was no significant difference in ruminal fluid pH levels among the three incubated treatments, indicating that the residual effects of MS and FS were minor and that aBC can nurture a suitable acid–base environment without resulting in rumen acidosis. The apparent discrepancy in the current results regarding rumen acidosis is

difficult to explain, but it cannot be ruled out that mixtures with MS or FS are safe for feeding ruminants. In fact, the current in vitro system utilizes a buffer and does not accurately represent what may occur in the rumen. Therefore, additional research is required to shed light on this issue.

The greater extent of substrate degradation resulted in an increased concentration of VFAs. VFAs are essential organic acids used by ruminant animals as energy sources. It is well-known that flavonoid-rich biomass could shift rumen VFA levels in goat rumen fluids [15]. This is because the hydroxyl groups in anthocyanins are the primary functional groups metabolized in the rumen through the hydrolysis of glycosides and cleavage of heterocyclic compounds. The main VFAs produced are acetate and butyrate. Our results could thus corroborate the results of previous studies [2,44,47] which reported that anthocyanin-rich plants or biomasses shifted the fermentation of VFAs by degrading substrate fermentation, resulting predominantly in the accumulation of acetic acid. In addition, our findings revealed no shifts in fermentation for propionic acid and butyric acid and this could explain the increase in total VFAs due to the increase in acetic acid contents as a consequence of higher concentrations of soluble carbohydrates, especially as a result of cellulose degradation. Consistent with our results, Hosoda et al. [44] and Tian et al. [47] reported that ruminal fluid acetic acid tended to increase after ruminant animals were fed anthocyanin-rich corn. This might be attributed to the involvement of anthocyanins in regulating the bacterial population, resulting in shifts in VFA production and other fermentation gases, such as the ruminal biogases methane and carbon dioxide, including alternative hydrogen sinks [20,21,47]. However, anthocyanin-rich plants or biomass were observed to have no effect on ruminal fluid VFA content [31]. The most probable reason for is the high concentrations of rumen-degradable anthocyanins. For example, anthocyanins from corn [44], purple corn stover silage [31], and purple corn [47] seemed to have higher degradation rates, but anthocyanins from colored barley [32] had low degradation rates in rumen fluid. Notably, optimization of VFA production could be achieved by matching the production requirements of the rumen host—the rumen microbiome and fermentation with the availability of fermentable substrates in rumen fluids being the most obvious influencing factors [48]. Overall, our data, together with previously published data, indicate that differences in the VFA production were due to anthocyanin sources, anthocyanin deliveries, fermentation substrates, and animal physiological stages.

A remarkable difference in the fermentation end-products mentioned above might be attributed to changes in the microbial population. Microbes shift a miscellaneous network in the rumen harbor and ferment fibers, including secondary compounds from plant material ingested by ruminants. A previous study demonstrated that when goats fed on dietary anthocyanins, anthocyanidins with one or more glucose, rhamnose, galactose, xylose, and arabinose as a constructor for flavonoid–anthocyanin could change a specific microorganism group and its numbers in the rumen harbor [47]. Similar to our study, incubated substrates rich in anthocyanin (PBC and silages treated with 4% MS + 0.030% FS) have been shown to have a positive impact on cellulolytic activity in relation to the modulation of prominent fiber-degraders in the rumen harbor. Our findings revealed that the abundances of *R. albus* and *R. flavefaciens* tended to increase in the PBC and PBC treated with 4% MS + 0.030% FS groups without changing ruminal fluid total bacteria compositions. Other predominant cellulolytic bacterial species for *F. succinogenes* found in the rumen were unaffected by PBC or silages treated with 4% MS + 0.030% FS or NA. Consistent with our results, Yusuf et al. [49] showed that *Andrographis paniculata* leaves rich in plant active substances (lactones, anthocyanin, flavonoids, sterols) in a goat diet had a tendency to increase the quantity of ruminal *R. albus* and *R. flavefaciens*, with the ruminal fluid total bacteria remaining unchanged, thereby improving the nutrient digestibility that could be achieved by goats. In our study, the average abundance of *R. albus* in incubated PBC and silages treated with 4% MS + 0.030% FS was approximately 4.44% higher compared with incubated SaCB1, whereas the average abundance of *R. flavefaciens* in incubated PBC and silages treated with 4% MS + 0.030% FS was approximately 17.32% higher compared

with incubated NA. In contrast, the average abundances of *R. albus* and *R. flavefaciens* in the rumen fluid of finishing steers was relatively unchanged, but the average abundance of *F. succinogenes* was relatively higher after intake of ensiled mulberry leaves or sun-dried mulberry fruit pomace rich in flavonoids and anthocyanins compared with normal mixed rations (without anthocyanins) [50]. The substrate fermentation used, the molecular weights of anthocyanins, feedstuff processing, and animal species might be responsible for this difference. Therefore, our findings demonstrated that the presence of higher levels of digestible carbohydrates and/or total soluble solids (especially glucose, sucrose, and fructose) in PBC and silages treated with 4% MS + 0.030% FS may have contributed to higher numbers of *R. albus* and *R. flavefaciens*, which probably resulted in differences in VFA concentrations (as shown above). However, most influences of the cellulolytic degraders were observed mainly in *R. albus* throughout 24 h of incubation, indicating that anthocyanins could modulate cellulolytic bacteria during cellulolytic activity in rumen harbors. Multi-omics analyses using a gnotobiotic sheep model confirmed that there was competition between *R. albus, R. flavefaciens,* and *F. succinogenes* due to adherence to and growth upon cellulosic biomass [51].

It is well-known that, sometimes, increases in total cellulolytic bacteria populations tend to decrease protozoa populations, as well as populations of methanogenic bacteria, after fermentation of polyphenol substrates, including flavonoid oligomers [21,22]. Similarly, our results somewhat resemble those of previous studies [49,50] which examined similar conditions; for instance, there was no significant effect on the abundances of protozoa in rumen fluids due to incubation of anthocyanin substrates. However, compared with the same study on quantifying the abundance of methanogenic bacteria, our results showed a reduction in methanogenic bacteria after incubation of aBC substrates (NA and silages treated with 4% MS + 0.030% FS). Along with non-cellulolytic contents low in rumen fluids, the lack of sifted protozoa might be due to the limited amounts of anthocyanins in PBC, NA, and silages treated with 4% MS + 0.030% FS themselves. Patra et al. [52] comprehensively reviewed studies of ruminal methanogens and responses with an emphasis on protozoa-associated methanogens. Protozoa-associated methanogens might not be related to *Methanobacterium* (e.g., methanogenic bacteria) due to the observation of smaller numbers of 16S *Methanobacterium* rRNA gene sequences recovered from protozoa—it is well known that methanogens are not associated with ruminal fungi and/or protozoa. Basically, plant flavonoid–anthocyanin had been shown to have a strong antibacterial activity by interrupting the semipermeable membrane of methanogens during intercellular interactions, but the efficacy of plant flavonoid-anthocyanin depended on the molecular weights, doses, type, sources, and basal substrate used [32,40,52]. Recently, iron-reducing bacteria have been extensively used to suppress methane producers, including methanogens, during anaerobic digestion [53]. The potential direct interspecies electron transfer between iron-reducing bacteria and methanogens with iron oxides could explain why the alleviation of methanogens occurred. Therefore, our results obtained by using FS as a source of residual iron in silages treated with 4% MS + 0.030% FS might have had a negative effect on free-living ruminal methanogens and therefore the treatment could be considered to have anti-methanogenic properties.

*4.4. The Incorporation of 4% MS + 0.030% FS into Silages Modulates the Sustainable Mitigation of the Ruminal Biogases Methane and Carbon Dioxide without Impairing Total Gas Production*

As hypothesized, both MS and FS modulated fermentable substrate gases in the headspace containing rumen culture at running time points. It is well-known that enteric ruminal gases, including hydrogen, methane, and carbon dioxide, are related to the abilities of rumen microorganisms to utilize cellulolytic and hemicellulolytic feedstuff during anaerobic fermentation degradation [48]. Notably, the degradation of plant material rich in cellulose and hemicellulose (soluble carbohydrates) in the rumen requires the colonization of ingested plant material by a complex microbial consortium and occurs in a time-dependent manner that is influenced by the nature of the substrate ingested [54]. In the present study, significant differences were not observed for ensiled anthocyanin-rich

black cane (aBC) treated with 4% MS and 0.030% FS incorporated into PBC with respect to carbon dioxide production and total gas accumulation, suggesting that the treatments produced similar total fermentable substrate patterns, such as total volatile fatty acid (VFA) content and addition to carbon dioxide yields. Not surprisingly, comparable total accumulated gas and carbon dioxide yields were achieved with incubated silages consisting of 4% MS + 0.030% FS and PBC due to their having relatively similar contents of carbohydrate and glucose in substrate fermentation. Our results were in agreement with previous studies [55,56] that investigated the gas production kinetics of some tropical and temperate forages with high quantities of nonfibrous carbohydrates and low lignin contents.

In the present study, incubated silage substrates consisting of 4% MS and 0.030% FS produced more total VFAs and acetic acid from greater quantities of available fermentable carbohydrates in rumen fluids, suggesting that a greater utilization of carbon dioxide and hydrogen by existing rumen acetogen populations via hydrogen-utilizing metabolic pathways might have occurred due to residual effects of FS. Reductive acetogenesis, reductions of carbon dioxide and hydrogen to acetate, could have thermodynamically outcompeted methanogenesis in the rumen [57], its functional predominance over methanogenesis being due to the fact that acetogens survive mainly by metabolizing carbohydrates [58]. Generally, methanogenesis is the primary biochemical pathway for the removal of metabolic hydrogen released through the fermentation of carbohydrates in the rumen, yielding methane gas, and one of the desirable strategies for enhancing methane mitigation by inhibiting methanogenesis uses a chemical compound and appropriate threshold of hydrogenotrophs that incorporate a hydrogen–electron sink pathway [57]. That said, the reduction of nitrate and sulphate thermodynamically outcompetes methanogenesis. Similarly, Greening et al. [59] reported that occurrences of nitrate and sulphate reduction and reductive acetogenesis in sheep rumens were due to the presence of genes and transcripts of hydrogenases catalyzing hydrogen uptake. Another study on the anaerobic treatment of sulphate-containing wastewater was reported [53]. Iron addition could directly inhibit methanogenesis because of the establishment of potential direct interspecies electron transfer between iron-reducing bacteria and methanogens with Fe oxides. Taken together, consistent with our results, residual sulphates combined with plant polyphenols effectively alleviate methane production by rumen cultures in vitro while not impairing feed digestion, fermentation, or microbial communities [10,52]. However, exceedances of the sulphate threshold (0.3–0.4% sulfur as sulphate) have been assessed as having a deteriorative effect on animal performance [52]. It appears from the available data that aBC containing 0.030% FS shows potential for methane mitigation, but further in vivo testing is required.

## 5. Conclusions

In summary, anthocyanin-rich black cane (aBC) is a grass rich in lignin and carbohydrates and could potentially be considered for use as ruminant feedstuff due to the abundance of soluble carbohydrates and anthocyanins it contains. The pre-ensiled treatment involving the addition of a mixture of 4% molasses and 0.030% ferrous sulphate to aBC resulted in an optimal balance of ensiling characteristics and is considered suitable for use in ruminants. Along with anthocyanin content in aBC, ferrous sulphate could be used as a rumen modifier for the following reasons: (1) fermentation enhancement; (2) control of rumen pH and nurturing of suitable acid–base conditions; and (3) regulation of rumen microbes by modulating fiber-degrading bacteria and inclusion of methane-producing bacteria. Therefore, the findings of this study could provide insights into hydrolytic pretreatments for lignocellulose agricultural biomass and a roughage-based strategy for the development of ruminant feedstuff. This would also have a significant positive impact on the environment. Further in vivo studies and feeding trials with ruminants and/or small ruminants are required to understand the mechanism by which anthocyanins in the absence or presence of residual ferrous sulphate and molasses affect digestibility, fermentation, antioxidant property, rumen microbes, and rumen-derived products.

**Supplementary Materials:** The following supporting information can be downloaded at: https://www.mdpi.com/article/10.3390/fermentation8060248/s1, Table S1: Primers used in the present study.

**Author Contributions:** Conceptualization, N.T.M.S., S.P., R.A.P.P. and P.P.; methodology, N.T.M.S., S.P., R.A.P.P. and P.P.; formal analysis, N.T.M.S., R.A.P.P. and P.P.; investigation, N.T.M.S. and R.A.P.P.; resources, N.T.M.S., R.A.P.P. and P.P.; data curation, N.T.M.S., R.A.P.P. and P.P.; writing—original draft preparation, S.P. and R.A.P.P.; writing—review and editing, N.T.M.S., S.P., R.A.P.P. and P.P.; visualization, N.T.M.S. and R.A.P.P.; supervision, S.P., R.A.P.P. and P.P.; project administration, N.T.M.S., S.P., R.A.P.P. and P.P.; funding acquisition, S.P., R.A.P.P. and P.P. All authors have read and agreed to the published version of the manuscript.

**Funding:** This research was funded by Suranaree University of Technology (SUT; contract no. Full-time 61/02/2021), Thailand Science Research and Innovation (TSRI), National Science Research and Innovation Fund (NSRF; project codes: 90464; 160368; FF3-303-65-36-17), National Research Council of Thailand (NRCT; project code: 900105), and Nakhon Ratchasima Rajabhat University (NRRU).

**Institutional Review Board Statement:** The Animal Ethics Committee of Suranaree University of Technology issued a statement approving the experimental protocol (SUT 4/2558). The research was carried out in accordance with regulations on animal experimentation and the Guidelines for the Use of Animals in Research as recommended by the National Research Council of Thailand (U1-02632-2559).

**Informed Consent Statement:** Not applicable.

**Data Availability Statement:** All data are contained within the article.

**Acknowledgments:** The authors would like to thank the staff of the Centre of Scientific and Technological Equipment and the the Suranaree University of Technology goat and sheep farm, Siriwan Phetsombat, and Thara Wongdee for the use of the research facilities and support in handling animals on the farm. Suong Ngo Thi Minh acknowledges a Suranaree University of Technology scholarship for ASEAN phase II (no.: MOE5636/320) as a funding source. The Figure 1 was created with BioRender.com (license no.: XV23Y8PBQD).

**Conflicts of Interest:** The authors declare no conflict of interest. The funders had no role in the design of the study; in the collection, analyses, or interpretation of data; in the writing of the manuscript, or in the decision to publish the results.

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
