# Peer review of "Optimizing Anthocyanin-Rich Black Cane (Saccharum sinensis Robx.) Silage for Ruminants Using Molasses and Iron Sulphate: A Sustainable Alternative"

_fermentation, doi:10.3390/fermentation8060248_

Round 1

Reviewer 1 Report

The have read the manuscript and recommended the following to authors:

1- revise the English with native speaker 

2- revise the statistical analysis and include in tables the p value of the effect of molasses (MS) and the effect of ferrous sulphate

Author Response

Reviewer #1

Revise the English with native speaker

Reply: We now revised thoroughly content in the manuscript and included the certificate of the English language required the assistance of a native speaker, please ask to the editor.

Revise the statistical analysis and include in tables the p value of the effect of molasses (MS) and the effect of ferrous sulphate

Reply: Thank you for reviewer’s suggestion, now the data were subjected to ANOVA using the general linear models (GLM) procedure of SAS 9.4 according to a 2×3 factorial arrangement in CRD instead of a 3×3 factorial arrangement in CRD (line 259).

Basically, those had been included in all tables. However, authors notice that our previous content still unclear. Therefore, we revised the significance p value of molasses (MS) and ferrous sulphate (FS) and included those values in tables.

Reviewer 2 Report

Review of Manuscript Fermentation-1698574

The aim of the manuscript was to evaluate the effect of adding molasses or/and iron sulphate on fermentation characteristics, microbial composition, and in vitro gas and methane production and ruminal fermentation characteristics of black cane. I would like to request the authors to consider the following major and specific comments and remarks:

Major comments

Abstract is not informative enough and does not reflect the content of the manuscript. Results are superficially described, and I would recommend showing some values to have an idea of the impact of the treatments.

The introduction is too general, and authors did not put efforts to really justify the realization of the experiment. In general, I could not see a structure and convincing story in the introduction. I am missing relevant literature that support the use of MS and FS for ensiling

The chapter M&M was properly described and with enough detail with some exceptions (see minor comments)

Results were well presented and described

Minor comments

L18-22: Abbreviations are complicated and difficult to follow or remember. Make them shorter and simple. Use e.g., only one letter for each word

L23: Is not clear what was the second trial about. Be clear

L24: Would not deteriorate rumen fermentation: what is meant with this? Not clear

L25-27: How where these parameters modulated? Better, worse, increment, decrements? Be clear

L27: Greatest silage production? How? This statement is not clear

L32: Recommendations? What about future studies? What are the implications for farmers and researcher?

L42-43: Not agree! Energy and protein content are most important as well productivity and adaptation to the region

L43-45: Reference?

L46: rumen-derived products? What do you mean?

L48: This means that the potential is low and contrast with statement in L42.

L55: Just by ensiling would not provide animals with highly nutritious feed! Rewrite

L57-73: Reduce this part. And I do not read a clear justification of using ferrous sulphate. Go into detail about some results presented in the literature to see how big the potential is

L75-85: It is not clear what can be the benefit of mixing molasses and FS. Be clear and find a comprehensible justification

L89-90: Ruminal fermentation profile? Based on which study/procedure?

L98: Rumen culture? What method is this? Use standard terms

L117: Which material was used? 60 or 120 d regrowth?

L121: I would not call this a “negative” control. Just control!

L123: That is not positive control. You have here your control treatment (cane without treatment) and the treated ones. Besides you took fresh cane to be analysed. The way here is described is confusing (L118-123)

L129: And what was the third factor? It seems to be a 2 x 3 factorial

L132: dilution rate?

L143: I am sure the syringes have not this name. This is the name of the method. Hohenheim Gas Test

L144: Were these dried for the in vitro incubations?

L163-165: These incubations times are not reliable for the procedure used and the equation used for the kinetic. Cumulative gas production must be measured in more time points

L166-169: the estimated values of the kinetic were not presented in any table. I assume these were not really estimated

L183: Fresh and pre-ensiled are the same

L280: the abbreviation was introduced before. Just use it

L282: The DM content is low. Please check!

L354: What is here meant with greater silage production?

L365-366: What is meant here with “fermentation profile was affected”? Clarify

L452: greater recovery than the fresh material is not possible because there are always loses. Rewrite the sentence

L541: Instead of using “saBC7” mention specifically the additives that were added here for better comprehension

L543-546: This statement is not admissible because the in vitro system work with a buffer and does not really reflect what could happen in the rumen

Author Response

Dear reviewer

We herewith attach our responses to your notes, please find the attached file within this submission.

Best regards,
Corresponding authors

Reviewer 3 Report

I found the paper very interesting; it was aimed to evaluate the best supplementation for ameliorate feeding utilization of a forage that is one of the main golas for the future animal nutrtion area of research. The authors very well descibed their experiment which was performed using adequate methodologies. The discussion was highly exaustive. In my opinion the paper is acceptable for the publication in present form.  

Author Response

Reviewer #3

Overall comments:

I found the paper very interesting; it was aimed to evaluate the best supplementation for ameliorate feeding utilization of a forage that is one of the main golas for the future animal nutrtion area of research. The authors very well descibed their experiment which was performed using adequate methodologies. The discussion was highly exaustive. In my opinion the paper is acceptable for the publication in present form.

Reply: Needless to say that we highly appreciate the reviewer’s qualification in that the work is acceptable for the publication in present form.

Round 2

Reviewer 1 Report

NO more comments, Thank you